# Interventions to Increase Physical Activity in Community-Dwelling Older Adults in Regional and Rural Areas: A Realist Synthesis Review Protocol

**DOI:** 10.3390/mps6020029

**Published:** 2023-03-12

**Authors:** Stephen Cousins, Rebecca McKechnie, Patricia Jackman, Geoff Middleton, Tshepo Rasekaba, Irene Blackberry

**Affiliations:** 1John Richards Centre for Rural Ageing Research, La Trobe University, Albury-Wodonga Campus, Wodonga, VIC 3689, Australia; 2School of Sport and Exercise Science, University of Lincoln, Lincoln LN6 7TS, UK

**Keywords:** realist review, behaviour change wheel, physical activity, older adults

## Abstract

The importance of physical activity (PA) for the health and wellbeing of older adults is well documented, yet many older adults are insufficiently active. This issue is more salient in regional and rural areas, where evidence of the most critical components of interventions that explain PA participation and maintenance in older populations is sparse. This realist review will (1) systematically identify and synthesise literature on PA interventions in community-dwelling older adults in regional and rural areas, and (2) explore *how* and *why* those interventions increase PA in that population. Using a realist synthesis framework and the behaviour change wheel (BCW), context–mechanism–outcome (C-M-O) patterns of PA interventions for older adults in regional and rural areas will be synthesised. Thematic analysis will be employed to compare, contrast, and refine emerging C-M-O patterns to understand how contextual factors trigger mechanisms that influence regional and rural community-dwelling older adults’ participation in PA interventions. This realist review will be the first to adopt a BCW analysis and a realist synthesis framework to explore PA interventions in community-dwelling older adults in regional and rural areas. This review will provide recommendations for evidence-based interventions to improve PA participation and adherence by revealing the important mechanisms apparent in this context. **Systematic review registration:** (PROSPERO CRD42023402499).

## 1. Introduction

By 2050, the global population of adults over 60 years of age will increase beyond 2 billion [1]. Population aging is further accentuated in regional and rural areas, with patterns of migration showing an out-migration of the younger population to metropolitan areas and an in-migration of older adults to regional and rural areas [2]. For instance, 35% of Australians aged ≥ 65-years live outside metropolitan centers [3] and 24.3% of English adults aged ≥ 65-years reside in rural locations [4].

Advancing age heightens the risk of developing chronic, degenerative health issues, including musculoskeletal, cardiovascular, and respiratory diseases, along with cancer, diabetes, cognitive decline, and multimorbidity [5,6]. These conditions lead to a loss of functional independence, capability, and quality of life in older adults [7,8]. Health disparities within older populations are also evident based on one’s location of residence. People living in regional and rural areas report poorer health outcomes and higher rates of morbidity [9], lower rates of leisure-time physical activity [10], as well as higher rates of obesity [11] and lower quality of life and social functioning [12] compared to their counterparts residing in metropolitan areas. These health disparities are likely associated with some of the unique challenges faced by older adults in regional and rural areas, such as reduced access to health care services (exacerbated by the challenges of transport and distance), fewer local amenities and infrastructure, health workforce shortages, and greater likelihood of suffering from social isolation and loneliness [4,13]. Consequently, there is a recognisable need to improve access to health care services and provide sustained health promotion opportunities for older adults living in regional and rural areas.

There is now considerable evidence demonstrating that regular and adequate participation in both active and passive forms of physical activity (PA) can help mitigate or manage health conditions in older adults and promote healthy aging [14,15]. Indeed, even a “low dose” of 75 min of moderate to vigorous intensity PA per week can improve health outcomes for those aged 60 years and above [16]. Engaging in regular PA provides additional benefits relevant to older adults, including numerous psychological benefits [17], improved cognitive function [18], and protective effects against the impacts of dementia and Alzheimer’s disease [19]. Similar positive effects have also been demonstrated in older adults in regional and rural community settings, with improvements in PA, physical function, and psychological indices reported [20]. Furthermore, PA participation can promote “social connectiveness” amongst older adults [21], which is particularly important for those living in regional and rural areas due to the greater risk of experiencing social isolation and loneliness because of their age and location. Although the potential benefits of PA are widely documented, older populations continue to be at greater risk of physical inactivity compared to middle-aged or younger adults [22]. For example, only 55% of men and 41% of women aged 75 years and above meet PA guideline recommendations in the UK [23], with a little over 25% of Australians aged over 65 years meeting their respective guidelines [24]. These contrast with rates of 61–82% among people aged 18–49 years [25]. 

Despite the well-known benefits of PA for healthy aging and the promotion of specific PA guidelines for older adults [26], there is little certainty of the most critical characteristics or components that explain the efficacy of PA interventions for older adults, including those living in regional and rural areas, especially in terms of *how* and *why* they are effective. Furthermore, researchers have also highlighted the need for greater understanding of the complex environmental and contextual factors underlying PA uptake and maintenance in older populations [27]. Therefore, given the lack of clarity surrounding the key determinants of effective PA interventions in community-dwelling older adults in rural and regional areas, research is required to advance understanding of the contextual factors and mechanisms underlying PA engagement in this population to better inform policy and practice. 

### Justification for This Review

Realist synthesis is a methodology that extends the scope of a traditional narrative or systematic review. Traditional systematic review approaches to evaluating interventions, such as meta-analysis, are generally employed to determine whether an intervention has been effective, and to what extent, but can lack explanatory power [28]. Given the complex, dynamic, and multi-faceted nature of interventions, it is important to explain the contextual factors and mechanisms associated with specific outcomes [29]. Thus, a review methodology that seeks to understand and unravel the underlying complexities of effective interventions is required to develop greater understanding of PA interventions that appear to be efficacious for community-dwelling older adults in rural and regional areas. 

Realist synthesis has emerged as a widely used strategy for understanding complex health and social interventions [30,31]. The purpose of a realist review is to go beyond examining intervention effectiveness to develop a fine-grained understanding of how an intervention works, for whom, and in what contexts [32]. Thus, by adopting a realist synthesis methodology, this approach offers an opportunity to uncover the mechanisms behind reported improvements in PA levels for older adults living in regional and rural areas. By doing so, this will clarify how improvements occur, who benefits exclusively, and what contexts (i.e., circumstances) are particularly important for interventions to be effective.

The objective of this review is to systematically identify and synthesise literature on PA interventions in community-dwelling older adults in regional and rural areas to explore how and why those interventions increase PA in that population. A realist synthesis will be conducted to address the following questions: (1) what are the contexts and mechanisms that increase physical activity among community-dwelling older adults in regional and rural areas; (2) what interventions are feasible, sustainable, and effective to be implemented for community-dwelling older adults residing in regional and rural areas; and (3) what support structures are required to improve the delivery of these interventions to this target population?

## 2. Experimental Design

In general, PA interventions are underpinned by assumptions of how they work to bring about their intended outcome(s). A realist review uses a systematic and theory-driven approach to refine these assumptions into theories, which can then be empirically tested. In a realist synthesis, the theory of how a program *“works”* is structured according to the *“context*–*mechanism*–*outcome”* (C-M-O) approach [32]. That is, the program theory is explained as the contextual (C) factor in addition to a potential resource mechanism (M_resource_), hypothesized to have triggered the relevant mechanism response (the underlying process or behavior) (M_response_) to generate the outcome of interest (O) [33]. The process of our realist review is focused on identifying, explaining, and testing these semi-predictable C-M-O patterns (called demi-regularities). For example, a theory could be proposed that for community-dwelling older adults residing in a regional or rural area (C), participating in a PA intervention that has a dedicated expert delivering the program who educates and encourages the participants (M_resource_) increases the participants’ confidence in their physical capabilities (M_response_), thereby facilitating regular participation in PA (O).

This realist review will be based on the approach of Pawson et al. [32] and will be consistent with publication standards for realist reviews (RAMESES criteria) [34]. The process will focus on identifying, explaining, and testing semi-predictable patterns or demi-regularities in C-M-O configurations. Stages will comprise literature search and screening, quality assessment, data extraction, data analysis and synthesis, and dissemination. An overview of the stages of the review is presented in Figure 1.

## 3. Procedure

### 3.1. Stage 1: Systematic Literature Search and Screening

A systematic literature search will be conducted to identify relevant studies of interventions aimed at increasing PA behaviours in community-dwelling older adults residing in regional and rural areas. Relevant literature will be obtained by the lead author (SC) by searching four electronic search databases: (1) CINAHL, (2) Embase, (3) Medline, and (4) SPORTDiscus. Systematic search strings will be designed using terms targeting three major constructs: (1) older adult, (2) physical activity/exercise/sport, and (3) community/regional/rural locations and will be combined with the AND operator. Similar key terms will be entered and separated by the term OR and truncation (*) used to capture all possible variations of selected key terms, as follows: (old* adult*/person*/people*/men*/women* OR old age OR aged OR aged, 80 and over OR senior* OR elder*), (physical activit* OR exercise* OR sport*), (rural* OR regional* OR community*/populat*/area*/town*/city*/location*). In addition to electronic database searches, the reference lists of included articles will be screened to identify any further articles that satisfy the eligibility criteria. 

Studies published up to July 2022, with no limit on earliest year of publication, will be included if they: (1) sample community-dwelling older adults (≥65 years), with or without diagnosed illness, living in regional/rural areas, regardless of previous experience or exposure to PA, exercise, or sport, of which at least 50% of participants are ≥65 years; (2) evaluate the outcomes of PA programs/interventions/behaviours, including active forms of intermittent and work-related physical activity (i.e., housework, gardening, manual labour), exercise, and sport; (3) contain original data; and (4) are published in the English language. Articles will not be restricted by country, study design, or outcome measures. Articles will be excluded if: (1) no outcomes of PA programs/interventions/behaviours are included; (2) the geographic location of the intervention (urban vs. rural) is not specified; (3) the intervention only includes passive forms of exercise and/or is rehabilitation- or treatment-focused; (4) the study is not published in English or is not an original investigation; and (5) <50% of participants are ≥65-years, or terms including *“seniors”*, *“elderly”*, or *“older adult”* are not used.

Identified article citations and abstracts will be uploaded to Covidence (SC), a web-based collaborative software platform that streamlines production of systematic and other literature reviews (Veritas Health Innovation, Melbourne, Australia—www.covidence.org, accessed on 1 June 2022). After removal of duplicates, articles will be screened in a two-stage process. Stages 1 and 2, respectively, will comprise title and abstract screening and full article review by two reviewers (SC and RM), with each article reviewed independently against the eligibility criteria by both reviewers. Articles will be accepted or rejected based on consensus. Discrepancies in decisions to include or exclude an article will be resolved by a third independent reviewer (TR), whereby the majority decision results in a study being included or excluded. A flow diagram of this systematic search process, recording excluded (with reasons) and included studies for data synthesis, will be included when presenting the results of this review. 

### 3.2. Stage 2: Quality Assessment

Each study will be assessed independently by two reviewers for rigour, risk of bias, and outcome quality, using the relevant Joanna Briggs Institute (JBI) appraisal tool based on the study design employed. JBI critical appraisal tools provide a systematic approach for assessing the methodological quality of a study and the extent to which it has addressed the potential for bias within its design, conduct, and analysis [35]. Results from each reviewer’s quality assessment will be discussed to assist with subsequent analysis, synthesis, and interpretation of findings.

### 3.3. Stage 3a: Data Extraction

Each article will undergo independent data extraction by any two reviewers from the research team. This will be followed by a team meeting during which the reviewer pair presents extracted data to the rest of the team, after which discussion and refinement of the extracted data will take place. Extracted data will include the following: authors’ details, year of publication, country, participant details including comorbidity and functional status, study design, and intervention details (setting, mode of delivery, follow-up, comparison groups), outcome measures, main findings (including physical activity outcomes), and hypothesised contextual components and mechanisms (how the intervention may have *“worked”* to trigger change). A copy of the data extraction form template is provided as Appendix A. Collating this information will allow for studies to be grouped in terms of similar participants, interventions, or setting characteristics as required during data analysis.

### 3.4. Stage 3b: Data Analysis and Synthesis

C-M-O configurations will be developed by each respective reviewer using the data extraction template, and will include:Context (intervention, setting, or participant characteristic to which a mechanism may be applied).Mechanism, resource (a strategy or approach applied to a given context).Mechanism, response (an intermediate outcome in direct response to the mechanism resource).Outcome (the desired/intended or measured output).

C-M-O patterns arising will be synthesised to develop program theories using the behaviour change wheel (BCW) [36] and realist synthesis framework [36]. Program theories will provide a structure for exploring the complex relationships between health interventions and outcomes, often involving diagrams or flow charts that convey the relationships between contextual factors, mechanisms, and outcomes [37]. 

In accordance with the realist synthesis approach [38], thematic analysis will be employed to compare, contrast, and refine emerging C-M-O configurations within and across studies to identify demi-regularities to understand how contextual factors trigger the mechanisms that influence community-dwelling older adults’ participation in PA programs. In addition, iterative content assessment between members of the research team, all of whom bring different theoretical and applied expertise, will stimulate critical discussions that assist in reaching consensus and refuting or refining the proposed C-M-O theories, whilst offering an opportunity to explore new theoretical propositions that can be generated from the data. Findings will be interpreted within the context of the BCW to assist in characterising interventions and linking intervention components to changes in PA behaviour.

### 3.5. Dissemination of Findings

Upon completion, findings of this review will be shared with academic and non-academic communities through peer-reviewed journal publications, conference presentations, presentations to relevant stakeholders and practitioners, and using the social media platforms of the authors’ affiliated institutions. It is important that the findings of the review, if indicating valuable information for health service delivery, are shared widely to enable the framework and recommendations for evidence-based interventions generated through this research to be translated into practice, with the ultimate goal of increasing PA in community-dwelling older adults in regional and rural areas. To the researchers’ knowledge, this review will also be the first to adopt a BCW analysis and realist synthesis framework to assess PA interventions in community-dwelling older adults in regional and rural areas; therefore, the review process, as well as the review findings, will be included in any presentation or dissemination process. 

## 4. Discussion

### Limitations

Realist synthesis is not designed to report on the effectiveness of an intervention based solely on quantifiable outcomes, but rather is designed to contextualise success—or lack thereof—and the mechanisms that produce these outcomes. As such, frameworks for interventions may be developed, but definitive answers to research questions may not be achieved. Despite best efforts to describe the review process here, the iterative nature of a realist synthesis may see delineation from the review process as new themes are developed. As such, this BCW analysis and realist synthesis protocol is considered a flexible tool with which to initiate and guide the process, maintain the aim and scope, and establish the transparency of the process. Our adoption of a systematic literature search may increase reliance on published data above grey literature and, as such, this synthesis may be vulnerable to publication bias. Furthermore, by only including English-language publications, our review is also susceptible to language bias.

## 5. Summary

This protocol paper provides an account of our intended process in undertaking a realist synthesis and BCW analysis to understand which interventions assist in improving PA behaviours among community-dwelling older adults residing in rural and regional areas, why, amongst whom, and in what contexts. Existing evidence has previously been limited to older adult populations or rural and regional dwellers and has focussed on the effectiveness of programs, with limited insight into the contexts and mechanisms through which they may work. As such, the findings of this realist synthesis will provide rich insight into the mechanisms that underpin successful (or unsuccessful) interventions and the contexts in which they work. By doing so, this will extend our understanding of facilitators and barriers to implementing PA interventions among older adults in rural and regional areas and will ultimately provide relevant and timely recommendations for evidence-based interventions to improve PA uptake and potential health outcomes among this target population.

## Figures and Tables

**Figure 1 mps-06-00029-f001:**
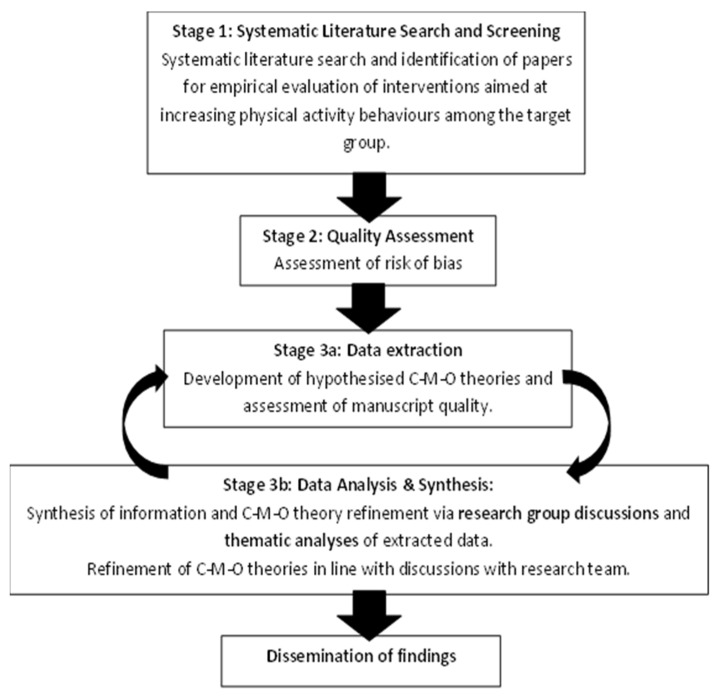
Schematic of proposed methodology.

## Data Availability

No new data were created or analyzed in this study. Data sharing is not applicable to this article.

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
