# Peer review of "Interventions to Increase Physical Activity in Community-Dwelling Older Adults in Regional and Rural Areas: A Realist Synthesis Review Protocol"

_mps, 2023, doi:10.3390/mps6020029_

Round 1
Reviewer 1 Report
Thank you for the opportunity to review the manuscript entitled, “Interventions to increase physical activity in community-swelling older adults in regional and rural areas: A realist synthesis review protocol”.
The authors described in this manuscript the protocol for a realist synthesis review with the goal to identify and synthesize literature on physical activity (PA) interventions in community-dwelling older adults in regional and rural areas, as well as to explore how and why such interventions increase PA in this population. The authors plan to synthesize context-mechanism-outcome (C-M-O) patterns of PA interventions for older adults in regional and rural areas using a realist synthesis framework and the behavioral change wheel (BCW). The authors will use thematic analysis to compare, contrast, and refine emerging C-M-O patterns to understand how contextual factors trigger mechanisms that influence this population’s participation in PA interventions. The authors propose that this review will provide recommendations for evidence-based interventions to improve PA participation and adherence by revealing important mechanisms in play.
The topic of investigation is of great relevance for a subgroup of older adults that often face health disparity. Overall this is a manuscript with strong potential. Background of the subject matter and rationale for the proposed methodology is well-written and cogent. Potential limitations considered and discussed. Additional information/considerations will strengthen the authors’ contribution and relevance for the audience.
PROCEDURE, Stage 1 – Please specify by whom the database searches will be performed (e.g. a research librarian).
PROCEDURE, Stage 1 – Given there is no restriction on study design, would the authors consider including Cochrane Library Online and PsycINFO as part of the search?
PROCEDURE, Stage 1 – Inclusion criteria states studies that have at least 25% of participants will be eligible. This invites the question in terms of representativeness of the study sample. In approaching studies that have mixed sample of older and younger adults, would suggest the authors to restrict to including studies that performed a subgroup analysis of older adults (aged ≥65) participants. Alternatively, the authors may consider increasing the 25% criterion to at least >50%.
PROCEDURE, Stage 1 – Studies will be excluded if the term “older adult” is not stated. Would suggest to rewrite this piece accordingly, to avoid potentially missing relevant articles that used the terms “seniors” or “elderly” etc.
PROCEDURE, Stage 3a – Please consider including comorbidity as part of the data extraction to provide more contextual information in addition to functional status, as it is known to be linked to physical activity.
Dissemination of findings: The authors may also consider using social media pages/channels of their respective affiliated institutions.
Thank you again for the opportunity to review this interesting and important manuscript.
Reviewer 2 Report
The manuscript entitled Interventions to Increase Physical Activity in Community Dwelling Older Adults in Regional and Rural Areas: A Realist Synthesis Review Protocol by Cousins et al. is a protocol document that summarizes the process strategy that the authors intend to follow, adopting a realist synthesis and behavioral change wheel analysis, to determine which interventions contribute to improving physical activity behaviors in community-dwelling and rural older adults. The authors' objectives are interesting, but some issues need to be addressed before approval.
Physical activity interventions significantly improve the quality of life of the elderly by reducing the rate of comorbidities and mortality. However, the elderly may not always engage in regular physical activity, as some of them may be forced into a sedentary lifestyle and therefore take advantage of alternative physical activity methodologies. One example is represented by whole body vibration, considered as a form of passive exercise, which has been able to reproduce the effects of physical activity both in the elderly and in mouse models (Bonanni R, Cariati I, Romagnoli C, D'Arcangelo G, Annino G, Tancredi V. Whole Body Vibration: A Valid Alternative Strategy to Exercise? J Funct Morphol Kinesiol. 2022 Nov 3;7(4):99. doi: 10.3390/jfmk7040099. PMID: 36412761; PMCID: PMC9680512)
In addition, the authors should be careful not to confuse physical activity with sports. Will the elderly who participate in sports from a young age be included in the study?
Will the results of the research strategy be summarized with a flow chart? This needs to be specified.
Although the authors aim to adopt a realist synthesis therefore not focusing on quantifying the outcomes of a particular physical activity intervention, it is important that the studies found in the literature are faithfully reported and that the conclusions are supported by the results. Therefore, I suggest detailing the results of the studies found in the literature, reporting the data and outcomes, and contextually providing an overview of the factors that determined the success or failure of a specific physical activity intervention, avoiding misleading conclusions and confusion to the reader. If the authors intend to adopt such a procedure, this should be reported in this manuscript.
